# A Rare Parasite in Cats: Record of a *Linguatula serrata* Frölich, 1789 (Porocephalida, Linguatulidae) Nymphal Infestation in a Cat in Albania, with a Synopsis and Review of the Literature on *L. serrata* Infestation in Cats

**DOI:** 10.3390/biology13121073

**Published:** 2024-12-20

**Authors:** Enstela Vokshi, Martin Knaus, Steffen Rehbein

**Affiliations:** 1Fakulteti i Mjekësisë Veterinare, Universiteti Bujqësor, Kodër Kamëz, 1000 Tirana, Albania; enstelashukullari@gmail.com; 2Boehringer Ingelheim Vetmedica GmbH, Kathrinenhof Research Center, 83101 Rohrdorf, Germany; martin.knaus@boehringer-ingelheim.com

**Keywords:** cats, ‘tongue worm’, *Linguatula serrata*, Pentastomida, nymph, lungs

## Abstract

*Linguatula serrata*, commonly known as the dogs’ ‘tongue-worm’, is a worm-like arthropod parasite of the class Pentastomida. It does not have five mouths as implied by the name, but there is one mouth flanked by two pairs of retractable hooks. *Linguatula serrata* has an indirect life cycle with canids as definitive hosts and herbivores as intermediate hosts which have a predator-prey relationship. Adult *L. serrata* usually live in the nose and pharynx of the definitive host, and the infectious larval stages (nymphs) are encapsulated in visceral organs of the intermediate host. However, domestic cats were listed in the literature to serve as both definitive and intermediate hosts of *L. serrata*. A thorough review of the literature including the herein described rare case of a nymphal infestation of a cat provides support for the potential role of the cat as (accidental) the intermediate host of *L. serrata*. There was, however, no support for cats to act as the definitive host of *L. serrata*.

## 1. Introduction

*Linguatula serrata* Frölich, 1789 (Porocephalida, Linguatulidae), also known as the ‘tongue worm’ or European *Linguatula*, is a cosmopolitan endoparasite, currently classified in the class Pentastomida within the phylum Arthropoda, with a two-host life cycle usually utilizing canids as its final hosts and chiefly various herbivorous mammals as intermediate hosts. Because the adult parasites inhabit the nasal cavities and sinuses of their hosts, potentially impair nasal respiration, and are occasionally expulsed from the nose, *L. serrata* infestations usually attract the attention of dog owners prompting the consultation of a veterinarian. In addition, *L. serrata* does have some zoonotic importance.

In several parts of the Middle East and Africa, *L. serrata* is highly endemic. Adult parasites are particularly prevalent in free-roaming dogs, and a large proportion of domestic ruminants carry the infectious larval stages of the parasite (‘nymphs’, ‘Pentastomum denticulatum’), which is called ‘visceral pentastomosis’. Human nasopharyngeal infestation acquired by the consumption of insufficiently cooked lymph nodes, liver, or other internal organs of *L. serrata*-infested ruminants causes a well-known clinical syndrome named halzoun or marrara. There are also several reports on human visceral pentastomosis, which is usually symptomless and associated with minimal pathology but may rarely, in the case of erratic nymphal migration, result in clinical presentations [1,2,3,4,5,6,7].

Throughout Europe, *L. serrata* infestation in animals was described in a rather limited number of publications in the second half of the 20th century. Local epidemiological data collected in dogs and wild carnivores were reported from Northern Ireland [8], former Yugoslavia [9,10,11,12,13,14,15], and Romania [16,17], and sporadic cases of infestation of dogs and/or wolves (*Canis lupus* Linnaeus, 1758) were described from Germany, Italy, and Greece [18,19,20,21]. In addition, visceral pentastomosis was reported in domestic ruminants from the British Isles, Spain, France, Italy, former Yugoslavia, Albania, Romania, Bulgaria, and Greece [17,22,23,24,25,26,27,28,29,30,31,32,33,34,35], and in hares (*Lepus europaeus* Pallas, 1778), southern chamois (*Rupicapra pyrenaica* Bonaparte, 1845), and/or European roe deer (*Capreolus capreolus* [Linnaeus, 1758]) from Spain, France, former Yugoslavia, Romania, and/or Bulgaria [16,36,37,38,39,40,41,42]. Furthermore, the diagnosis of *L. serrata* in dogs imported into Germany from Turkey was reported [43]. A substantial number of publications were issued in the first decades of the 21st century comprising case reports on the finding of *L. serrata* in dogs translocated from Romania but also from Spain, Italy, and Turkey into the UK, Germany, Norway, Finland, and Italy [44,45,46,47,48,49,50,51,52,53,54]; there are also records of *L. serrata* infestation in local dogs in the UK, Spain, Italy, Serbia, Albania, Romania, Bulgaria, and Greece [55,56,57,58,59,60,61,62,63,64,65,66,67,68] and in wild canids (wolf, golden jackal [*Canis aureus* Linnaeus, 1758] and/or red fox [*Vulpes vulpes* (Linnaeus, 1758)]) in Italy, Bosnia and Herzegovina, North Macedonia, Romania, and Greece [69,70,71,72,73,74,75]. In addition, there are records of the occurrence of visceral infestation with *L. serrata* nymphs in hares and/or roe deer from Romania, Bulgaria, and Greece [74,76,77,78,79,80,81], while reports on *L. serrata* nymphal infestation of domestic ruminants, originating from Italy, Romania, Bulgaria, and Greece, are rather scarce [76,82,83,84].

Overall, the publications mentioning *L. serrata* infestation in animals in Europe since the 1950s identify southeastern Europe, in particular the Balkans and Romania, as the geographic focus of the endemicity of *L. serrata*. After a decline in the number of reports issued on *L. serrata* infestation from the 1950s to the 1970s, an increase can be observed towards the end of the century continuing into the first decades of the 21st century. This trend is driven by the growing number of reports on *L. serrata* infestation in relocated rescue dogs and in dogs living in endemic regions. In addition, there is an increasing number of publications which report the results of research in wildlife species, indicating the existence of a sylvatic cycle and a wildlife reservoir of *L. serrata* in Europe in addition to the circulation of the parasite among domesticated animals.

While the role of canids in the transmission cycle of *L. serrata* is well defined, the position of felids and especially that of domestic cats appears to be not that clear. Interestingly, the comprehensive textbook on feline parasitology of Bowman et al. [85] does not notice *L. serrata* as a parasite of cats at all but refers only to nymphal infestations with the pentastomid *Armillifer armillatus* whose adults reside in the respiratory tract of large snakes. There are publications which list cats as both final and intermediate hosts of *L. serrata* [86] or mention cats as final hosts of *L. serrata* (among others, [54,67,87,88,89]) or consider felids generally as non-competent final hosts of *L. serrata* [2]. None of these publications provide support for cats acting as final hosts for the parasite or the inadequacy of cats to serve as final hosts. However, there have been several reports which describe the infestation of cats with nymphs of *L. serrata*, indicating that this parasite is worthy of consideration. Interestingly, the first finding of *L. serrata* nymphs in a cat [90] was reported just 40 years after the initial description of the parasite from a hare by Frölich in 1789 [91].

Given the limited knowledge and controversial positions on the role of cats as hosts of the pentastomid *L. serrata* [92], this communication reports a case of a *L. serrata* nymphal infestation identified in a domestic cat from Albania and aims to provide a synopsis and review of the literature on *L. serrata* infestation in cats.

## 2. Case Presentation

In the context of the examination of helminth parasites of domestic cats from the greater Tirana area [93], the lungs of 73 cats were available for the examination of lungworms. For lungworm recovery and count, the airways of the lungs were opened lengthwise using fine scissors, and helminths found in the trachea and bronchi were collected. In order to recover lung parenchyma dwelling small lungworms, the lung tissue was thereafter subjected to peptic digestion, and the digest material was examined under a dissecting microscope [94].

In the lung tissue digest material of an approximately one-year-old male mixed-breed cat examined in July 2010, a nymphal pentastome was revealed and collected. The specimen was mounted in lactophenol for closer examination of its morphology (body shape, hooks, number of cuticular annuli, annular spines) by light microscopy (Figure 1 and Figure 2). The nymph had 73 cuticular annuli, a body length of 4.3 mm, and a maximum body width of 1.2 mm. Two pairs of single hooks were present ventrally on the anterior part of the nymph flanking the mouth opening. Based on the morphological characteristics, which corresponded to the descriptions of nymphs of *L. serrata* [42,81,95,96], the specimen isolated from the lungs of the cat was identified as a nymph of *L. serrata* tongue worm.

For completeness, apart from the 1 *L. serrata* nymph, 64 *Aelurostrongylus abstrusus* (Railliet, 1898) and 2 *Eucoleus aerophilus* (Creplin, 1839) were recovered from the lungs of the cat, 8 *Toxocara cati* (Schrank, 1788), 34 *Ancylostoma tubaeforme* (Zeder, 1800), and 41 *Dipylidium caninum* Linnaeus, 1758 from the gastrointestinal tract, and 17 *Pearsonema plica* (Rudolphi, 1809) from the urinary bladder.

## 3. Discussion

This case presents the first report of the infestation of a cat with *L. serrata* (nymphal stage) in Albania and adds to the overall limited number of reports on the infestation of cats with *L. serrata* reported from Europe, Asia, and South America (Table 1).

Further to the reports summarized in Table 1, a comprehensive review of the parasites of dogs and cats in Austria, which claims to be based on ‘… data documented in the laboratory logbooks of the institute [unpublished], own studies and relevant literature [with references going back to 1852] …’, lists *L. serrata* as a parasite of both dogs and cats [107]. Unfortunately, neither the studies of the authors nor the referenced literature gave support to the occurrence of *L. serrata* in Austria. In addition, there is a more recent publication from Egypt reporting the results of the examination of 113 bulk fecal samples buried by stray cats [108]. Examination of the material included gross visual inspection for cestode segments and nematode specimens followed by a centrifugal flotation and identification of parasite eggs and cysts based on their morphology. Apart from fecal stages of various protozoans and helminths, the finding of (*loc. cit.*) “… two arthropod species, *L. serrata* (2%) and mites eggs (13%)” was reported. This finding was noted in that paper one more time, (*loc. cit.*): “Mite eggs and sometimes mites larvae were found in 13% of examined fecal samples as well *L. serrata* larvae were identified only in two samples (2%)”. Unfortunately, the record of *L. serrata* was not discussed at all, and the *L. serrata* findings are inconclusive because it remains unclear which technique was used and which *Linguatula* stages have actually been observed and/or recovered—*Linguatula* eggs (which contain the first larval stage) or ‘free’ *Linguatula* larvae (nymphs, which passaged the gastrointestinal tract following oral ingestion of nymph-containing organs of intermediate hosts). To the best knowledge of the authors, there is no evidence that adult *L. serrata* have ever been isolated from domestic cats, and cats have never been reported to shed *Linguatula* eggs. However, considering that *L. serrata* can be assumed to be abundant in Egypt [89,109,110,111], it may theoretically be imaginable that the fecal material of the cats has been accidentally contaminated by *Linguatula* eggs containing nasal secretions of dogs harboring adult tongue worms. A passage of nymphs through the gastrointestinal tract may potentially be possible given that the nymph recovered from the cat in the case reported here was isolated by the peptic digestion of the lung tissue, and peptic digestion was used repeatedly for the recovery of *L. serrata* nymphs from the tissues of ruminants [112,113]. It is, of course, unclear whether pentastomid nymphs will withstand the passage of the entire gastrointestinal tract of cats in a state allowing their recovery from the feces and appropriate identification. The recovery of *L. serrata* nymphs from feces, after passing the entire gastrointestinal tract, by standard flotation techniques seems rather unlikely. A collection of the nymphs during the gross examination of the feces may be excluded because the author specifically described cestode segments and nematodes targeted by this procedure.

The cat described in this report originated from the greater Tirana area and was examined around the same time when 1 of 602 dogs included in a survey to establish the endoparasite status of client-owned dogs was found shedding *L. serrata* eggs [63]. The questionnaire for this dog—a working dog from a rural habitat—revealed raw meat/offal feeding as a potential risk factor for contracting *L. serrata*. Apart from that record, the authors are not aware of any other report on the occurrence of *L. serrata* in dogs in Albania. However, abattoir-slaughtered cattle, sheep, and goats in Albania have previously been reported to harbor *L. serrata* nymphs [23], suggesting that a domestic cycle of the transmission of the parasite occurs in the country.

Most *L. serrata* nymphs in cats were isolated from the lungs and liver, but pleura, peritoneum, and intestine were reported also as locations of the finding of the nymphs (Table 1), indicating that nymphal development in cats may occur throughout the chest and abdominal cavities with the preference of the lungs and liver. The nymphs were found encapsulated in thin-walled, whitish tissue capsules located at the surface of the organs (Table 1). In the case reported here, the nymph was recovered following digestion of the lung tissue. It is therefore not clear whether the nymph was originally encapsulated at the surface of the cat’s lungs or whether the nymph was located within the lung parenchyma. The recovery of *L. serrata* nymphs from the lung parenchyma has been reported in the past from a dog, but also from hares and rabbits [41,77,81,96,114,115].

The level of infestation in the case reported here with one nymph was very low, as reported in the majority of the cases described in the literature (see Table 1). However, a maximum count of 36 and 14 nymphs recovered from the liver and the lungs, respectively, has been reported from one cat from Croatia [99]. That, in most cases, reference was made to a superficial (subserosal) localization of the encapsulated nymphs may suggest that the figures as to the level of infestation constitute minimum counts because the nymphs encapsulated in the parenchyma of the visceral organs may have been missed as there is only minimal pathology associated with low-level infestation. None of the reports summarized in Table 1 made any reference to specific pathology or clinical signs in the cats, suggesting that the visceral nymphal infestations were unsuspicious, and the recovery of the nymphs during the necropsy of the cats was considered as an additional or accidental finding. However, it may be worth referring to the clinical cases of pulmonary *L. serrata* pentastomosis reported in heavily infested hares and rabbits [41,115].

The shape of the nymph recovered from the lungs of the cat resembled that of the adult vermiform pentastome, and its size corresponded with measurements obtained from *L. serrata* nymphs collected from the lungs and liver of cats [99,105,106] and nymphs isolated from various herbivorous species [17,41,81,83,95,96,116].

While the recording of *L. serrata* nymphs implicates that domestic cats may, in principle, serve as intermediate hosts in the life cycle of *L. serrata*, the authors did not find any information in the literature on the recovery of adult parasites from cats which would support a role as a final or definitive host of *L. serrata* for domestic cats. There is, however, experimental work which supports that domestic cats are not suitable final hosts for *L. serrata* [117,118]. The authors [117,118] inoculated 10 domestic cats orally with various numbers of *L. serrata* nymphs collected from lymph nodes of sheep and goats, observed the animals for clinical signs, and examined them in intervals from 5 to 60 days after inoculation for the recovery of the parasites. Within a few hours after inoculation, the cats began to sneeze, cough, and shake their heads, with two cats showing sneezing and coughing episodes until necropsy 7 and 14 days after inoculation. *Linguatula serrata* nymphs, few in total, were recovered mainly from the nasal cavity of the cats up to 14 days post-inoculation; five cats examined from 32 to 60 days after inoculation were completely negative. In this context, it is worth mentioning that the examination of the heads of more than 1500 wild carnivores in Romania, including wolf (n = 407), red fox (n = 712), wild cat (*Felis silvestris* Schreber, 1777) (n = 88), lynx (*Lynx lynx* [Linnaeus, 1758]) (n = 30), badger (*Meles meles* [Linnaeus, 1758]) (n = 65), martens (*Martes* spp.) (n = 68), least weasel (*Mustela nivalis* Linnaeus, 1758) (n = 15), stoat (*Mustela erminea* Linnaeus, 1758) (n = 16), polecat (*Mustela putorius* Linnaeus, 1758) (n = 106), and raccoon dog (*Nyctereutes procyonides* [Gray, 1834]) (n = 7), revealed adult *L. serrata* only in the canids wolf (64.3%) and red fox (2.2%), as well as in the felid lynx (6.6%) [16]. As a potential reason for the absence of tongue worms in the other species, the authors discuss that the morphology of the upper respiratory tract, especially the smaller dimensions of the nasal turbinates and cavities in these species compared to that in the larger carnivores, does not provide the appropriate conditions for the development of *L. serrata* nymphs into adult parasites [16].

## 4. Conclusions

This case of a nymphal *L. serrata* infestation in a cat from Albania adds to the reports on the finding of this parasite and stage in domestic cats which are documented from several countries worldwide. The infestation of cats appears to be related to the presence of *L. serrata* cycling in domestic dogs and domestic herbivores as the main definitive host and main intermediate hosts, respectively, and seems to usually be subclinical in nature. Like the herbivorous main intermediate hosts of *L. serrata*, cats may become infested by the accidental ingestion of the larvated (infectious) *Linguatula* eggs discharged by a final host with nasal secretions or feces and contaminating feed and/or water. Ingested *L. serrata* eggs may develop in the visceral organs of cats into nymphs that constitute the infectious stage for the final host. Cats may therefore serve as accidental intermediate hosts for *L. serrata*. However, cats are unlikely to be of epidemiological relevance in the parasite’s transmission cycle as cats are not known to constitute an important prey of the known definitive hosts. Importantly, the review of the literature did not reveal any evidential information to support that domestic cats can act as final hosts of *L. serrata*.

## Figures and Tables

**Figure 1 biology-13-01073-f001:**
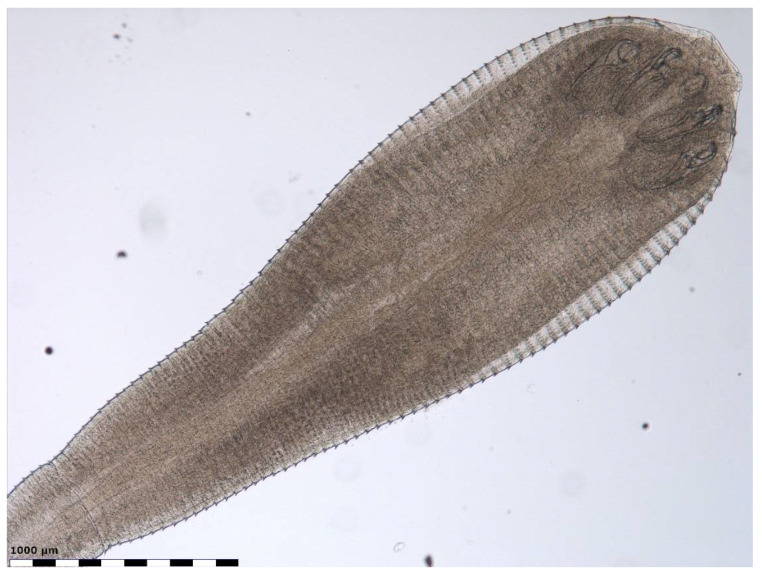
Nymph of *Linguatula serrata* collected from the lungs of a cat.

**Figure 2 biology-13-01073-f002:**
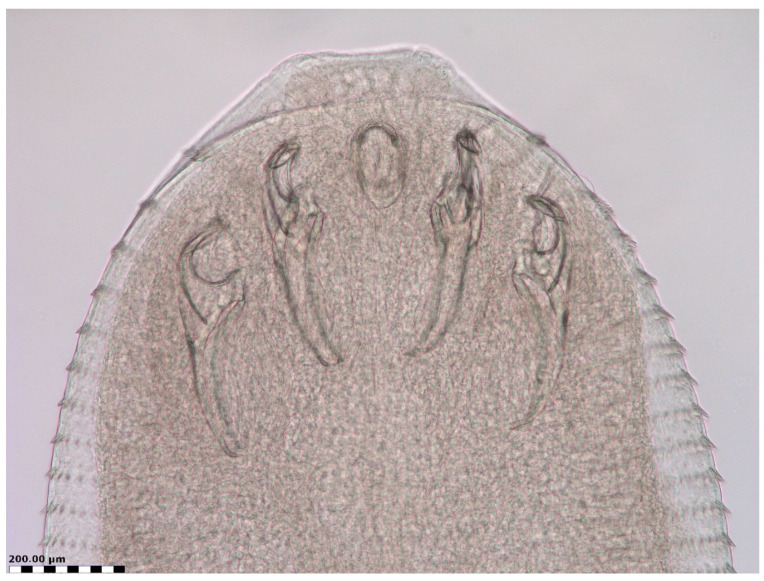
Anterior end of nymph of *Linguatula serrata* collected from the lungs of a cat, with central mouth opening flanked by two pairs of hooks (ventral view).

**Table 1 biology-13-01073-t001:** Synopsis of publications reporting on the finding of *Linguatula serrata* nymphs (visceral pentastomosis) in domestic cats.

**Country of Study**	**Time/Period of Study**	**Necropsy Type**	**Findings, Examination Techniques**	**Number of Nymph-Positive Cats/Number of Cats Examined**	**Number of Nymphs** **Isolated per Cat**	**Reference**
Germany	1824	No information	Gross examination, microscopy; small whitish cyst at liver surface with one nymph	1/Unknown (case report)	2 (liver)	[90]
	No information	Routine necropsy	Gross examination, microscopy; two small cysts at surface of the lungs, one calcified; one nymph per cyst	1/Unknown (case report)	2 (lungs)	[97]
France	No information	Parasitological Necropsy ^1^	Gross examination, microscopy; one cat with one cyst at pleura, one cat with one cyst at peritoneum, three cats with cysts (1 to 5) at lungs’ surface	5/76	1 (pleura), 1 (peritoneum), 1 to 5 (lungs)	[98]
Croatia	No information (likely close prior to report)	Parasitological necropsy	Gross examination, microscopy; one cat with cysts at liver + lungs’ surface (36 and 14, respectively), one cat with cysts (2) at liver surface; one nymph per cyst	2/100	2 and 36 (liver), 14 (lungs)	[99]
Greece	01/1974–09/1977	Parasitological necropsy	Gross examination, microscopy; nymphs collected from the surface of the liver of three cats	3/123	No information (liver)	[100] ^2^
Albania	2008–2010	Parasitological necropsy	Peptic digestion of the total lung tissue after opening of the lungs’ air passages, microscopy; one nymph isolated from the tissue digest of one cat	1/73	1 (lungs)	This report
Turkey	05/1981– 01/1984	Parasitological necropsy	Gross examination, microscopy; two cats with one cyst per cat; one nymph per cyst	2/100	1 (intestine), 1 (liver)	[101]
	1995–1997	Parasitological necropsy	Gross examination, microscopy; one cat with one cyst at liver surface containing one nymph	1/100	1 (liver)	[102]
Iran	No information (likely close to prior report)	Routine necropsy	Gross examination, microscopy, histopathology; one cat with few white, fine nodules at lungs’ surface	1/Unknown (case report)	“few” (lungs)	[103]
Bangladesh	07/2006–06/2007	Parasitological necropsy (only respiratory tract)	Gross examination, microscopy, histopathology; five cats with nymphs	5/36	1 to 4 (“… larynx to lungs …”)	[104]
Peru	1968	Routine necropsy	Gross examination, microscopy; one cat with two nymphs in the lungs	1/Unknown (case report)	2 (lungs)	[105]
Chile	03/1970–12/1971	Parasitological necropsy	Gross examination, microscopy; nymphs in the lungs, liver, and lungs + liver of eight, three, and one cats, respectively	12/100	1 to 4 (1, 2, 3, and 4 nymphs isolated from 7, 3, 1 and 1 cat, respectively)	[106]

^1^ Parasitological necropsy (includes routine examination of the gastrointestinal tract with liver, heart, respiratory tract, and urinary tract for parasites). ^2^ Information on organ and period of study provided by E. Papadopoulos after reviewing the thesis of S. T. Haralampidis (in Greek) which was the basis for the summary published in 1987 (personal communication, 18 June 2024).

## Data Availability

Not applicable.

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
