# Peer review of "A Rare Parasite in Cats: Record of a Linguatula serrata Frölich, 1789 (Porocephalida, Linguatulidae) Nymphal Infestation in a Cat in Albania, with a Synopsis and Review of the Literature on L. serrata Infestation in Cats"

_biology, 2024, doi:10.3390/biology13121073_

Round 1

Reviewer 1 Report

Comments and Suggestions for Authors

The manuscript is a short report on the first finding of the Linguatula serrata nymph in a cat from Albania. The work is a good addition to our knowledge about distribution and hosts of L. serrata. The Сommunication is well- written and structured; its theme is relevant to Biology. Introduction contains good review of the literature on research topic. It is worth noting the wonderful original drawings of the parasite.

However, I have some remarks about this manuscript.

In Abstract (last sentence) “…final or definitive hosts…”. These mean the same thing. Please, use one term here.

The morphological description of the L. serrata nymph says nothing about hooks and a mouth opening. Need more details…

Page 3 – Capillaria aerophila and Capillaria plica are obsolete species names. This species is now called Eucoleus aerophilus (Creplin, 1839) and Pearsonema plica (Rudolphi, 1819), respectively.

Note 2 on Table 1 – This should be formatted as a Personal Communication in the References.

The penultimate sentence in the abstract – Here domestic cats can be called an abortive host. This can also be mentioned in the Conclusion.

According International Code of Zoological Nomenclature (ICZN) at the first mention of genera or species in the article text its full Latin name with the author and year of description should be given; in relation all species of living organisms (In Introduction: Linguatula serrata Frölich, 1789; Page 3 – Aelurostrongylus abstrusus (Railliet, 1898), Eucoleus aerophilus (Creplin, 1839), Toxocara cati Schrank, 1788, Pearsonema plica (Rudolphi, 1819), etc.

And for the main “hero” of the article, please, add the names of higher taxa: “Linguatula serrata Frölich, 1789 (Porocephalida, Linguatulidae) …”

In the title of the Article, it is also necessary to add higher taxa.

Small recommendation: In scientific articles, it is better to use not the usual names of animals, but their Latin names (For example: Canis lupus Linnaeus, 1758 instead of wolf, Canis aureus Linnaeus, 1758 instead golden jackal, Vulpes vulpes (Linnaeus, 1758) instead of red fox, etc.)

The manuscript can be published in the Biology, but minor corrections are needed.

Author Response

Reviewer 1

Comments and Suggestions for Authors
The manuscript is a short report on the first finding of the Linguatula serrata nymph in a cat from Albania. The work is a good addition to our knowledge about distribution and hosts of L. serrata. The Сommunication is well- written and structured; its theme is relevant to Biology. Introduction contains good review of the literature on research topic. It is worth noting the wonderful original drawings of the parasite.
However, I have some remarks about this manuscript.
The authors wish to thank the reviewer for the positive consideration of the manuscript acknowledging specifically the value of the review of the literature which is not easily accessible and provides insight into many species which have been reported as hosts of Linguatula.

In Abstract (last sentence) “…final or definitive hosts…”. These mean the same thing. Please, use one term here.
Addressed.

The morphological description of the L. serrata nymph says nothing about hooks and a mouth opening. Need more details…
Information added.

Page 3 – Capillaria aerophila and Capillaria plica are obsolete species names. This species is now called Eucoleus aerophilus (Creplin, 1839) and Pearsonema plica (Rudolphi, 1819), respectively.

Note 2 on Table 1 – This should be formatted as a Personal Communication in the References.
Wording of Note 2 on Table 2 was changed in that “in litt.” was changed to “personal communication”. However, as a personal communication is not considered as a reference in the actual sense, the personal communication was not added to “References”.

The penultimate sentence in the abstract – Here domestic cats can be called an abortive host. This can also be mentioned in the Conclusion.
This comment is acknowledged. However, the authors are not aware about the category “abortive host” related to the life cycle of parasites. Therefore, the authors wish not to introduce this term in the manuscript but keep characterizing the domestic cat as accidental intermediate host.   

According International Code of Zoological Nomenclature (ICZN) at the first mention of genera or species in the article text its full Latin name with the author and year of description should be given; in relation all species of living organisms (In Introduction: Linguatula serrata Frölich, 1789; Page 3 – Aelurostrongylus abstrusus (Railliet, 1898), Eucoleus aerophilus (Creplin, 1839), Toxocara cati Schrank, 1788, Pearsonema plica (Rudolphi, 1819), etc.
Done.

And for the main “hero” of the article, please, add the names of higher taxa: “Linguatula serrata Frölich, 1789 (Porocephalida, Linguatulidae) …”
In the title of the Article, it is also necessary to add higher taxa.
Done.

Small recommendation: In scientific articles, it is better to use not the usual names of animals, but their Latin names (For example: Canis lupus Linnaeus, 1758 instead of wolf, Canis aureus Linnaeus, 1758 instead golden jackal, Vulpes vulpes (Linnaeus, 1758) instead of red fox, etc.)
The comment is acknowledged, and Latin names were added to the usual names of non-domesticated species names when appearing for the first time in the text. 

The manuscript can be published in the Biology, but minor corrections are needed.

Reviewer 2 Report

Comments and Suggestions for Authors

Excellent work, congratulations

Some minor comments:

As veterinarians, the authors should stick to the recommendations of the World Federation for the Advancements of Veterinary Parasitology and should avoid disease names ending on –iasis. Thus, several times the authors used pentastomiasis. This should be changed to pentastomosis.

Unfortunately, at the beginning of the text line numbers are missing. In the last paragraph of the case presentation, the reviewer would suggest to omit nematodes after Capillaria aerophila, ascarids after Toxocara cati, hookworms after Aelurostrongylus tubaeforme, tapeworms after Dipylidium caninum and nematodes after Capillaria plica.

In tab. 1: the 5th column; maybe it is better to write no. pos/ No. examined.

Line 3: the institute is most probably Institut fuer Parasitologie und allgemeine Zoologie of the Wiener Tieraerztliche Universitaet.

Line 23-25: maybe the author has accidently included a dog faecal sample when he examined cat faecal samples since he collected the samples from the environment. Also it is possible that the cat has eaten a lizard infected with pentastomids, for example a Railietiella sp. and eggs were passed with the faeces.

Author Response

Reviewer 2

Comments and Suggestions for Authors
Excellent work, congratulations
Some minor comments:
As veterinarians, the authors should stick to the recommendations of the World Federation for the Advancements of Veterinary Parasitology and should avoid disease names ending on –iasis. Thus, several times the authors used pentastomiasis. This should be changed to pentastomosis.
Addressed.

Unfortunately, at the beginning of the text line numbers are missing. In the last paragraph of the case presentation, the reviewer would suggest to omit nematodes after Capillaria aerophila, ascarids after Toxocara cati, hookworms after Aelurostrongylus tubaeforme, tapeworms after Dipylidium caninum and nematodes after Capillaria plica.
Deleted as recommended.

In tab. 1: the 5th column; maybe it is better to write no. pos/ No. examined.
Changed as recommended.

Line 3: the institute is most probably Institut fuer Parasitologie und allgemeine Zoologie of the Wiener Tieraerztliche Universitaet.
This comment is acknowledged. The respective paper was from authors working at that institute as given in the affiliations. However, the text passage referred to in the manuscript does only say “institute”. Therefore, the text of the manuscript was not changed.

Line 23-25: maybe the author has accidently included a dog faecal sample when he examined cat faecal samples since he collected the samples from the environment. Also it is possible that the cat has eaten a lizard infected with pentastomids, for example a Railietiella sp. and eggs were passed with the faeces.
This comment is acknowledged. However, it is felt that the description provided already sufficiently indicates that – whatever the stages seen were in fact – they were unlikely stages which derived from a pentastomid residing in the cat.

Reviewer 3 Report

Comments and Suggestions for Authors

The authors report a case of a L. serrata nymphal infestation identified in a domestic cat from Albania and aimed to provide a synopsis and review of the literature on L. serrata infestation in cats. The authors have tried to bring out the appropriate literature to support their case. This is novel and commendable work and could even be made better with the following:-

1. L. serrata is generally known as a parasite of carnivores of which the cat is one. Where is the novelty in this case?

2. The following statement about L. serrata recovered from cats in Egypt, "L. serrata findings are inconclusive because it remains unclear which technique was used and which Linguatula stages have been actually observed and/or recovered" seems to cast doubt on the authenticity of the results of the researchers in order to promote your own. Do you doubt the evidence presented in that publication?

3. The language though intelligible needs to be further edited

Author Response

Reviewer 3

Comments and Suggestions for Authors
The authors report a case of a L. serrata nymphal infestation identified in a domestic cat from Albania and aimed to provide a synopsis and review of the literature on L. serrata infestation in cats. The authors have tried to bring out the appropriate literature to support their case. This is novel and commendable work and could even be made better with the following:-
1. L. serrata is generally known as a parasite of carnivores of which the cat is one. Where is the novelty in this case?
The manuscript does not claim to provide the first description of Linguatula serrata parasitism in cats. However, the description of this rare case of parasitism in cats was taken to review the respective literature and, based on this, to clarify that any reference made that the cat may be a definitive host of L. serrata is not supported in the literature.

2. The following statement about L. serrata recovered from cats in Egypt, "L. serrata findings are inconclusive because it remains unclear which technique was used and which Linguatula stages have been actually observed and/or recovered" seems to cast doubt on the authenticity of the results of the researchers in order to promote your own. Do you doubt the evidence presented in that publication?
As explained in some detail in the manuscript, these findings, based on the examination of feces, are lacking clarity and tracebility and are thus questionable or doubful for several reasons. This does not interfere at all with the description of the own case which is based on the recovery of a nymph from a cat following necropsy. However, as indicated above, the intention is to raise attention.

3. The language though intelligible needs to be further edited.
This comment is acknowledged.